# Bioinspired Artificial Muscles Based on Sodium Alginate-Wrapped Multi-Walled Carbon Nanotubes and Molybdenum Disulfide Composite Electrode Membrane

**DOI:** 10.3390/polym15173535

**Published:** 2023-08-25

**Authors:** Yingxin Ji, Keyi Wang, Gang Zhao

**Affiliations:** College of Mechanical and Electrical Engineering, Harbin Engineering University, Harbin 150001, China

**Keywords:** biomimetic artificial muscles, sodium alginate, electrode membrane, MoS_2_

## Abstract

Using a naturally extracted polymer sodium alginate extracted from natural seaweed as the primary raw material, we have successfully developed an electroactive actuator known as biomimetic artificial muscle (BMAM). In comparison to conventional synthetic materials, this BMAM aligns more coherently with the prevailing principles of environmentally friendly development. During the preparation of the BMAM electrode membrane, we employed ultrasonic oscillation to adsorb varying quantities of MoS_2_ onto a reticulated structure formed by multi-walled carbon nanotubes (MWCNTs), thus enhancing the mechanical and electrochemical performance of the BMAM. Scanning electron microscopy and energy-dispersive X-ray spectroscopy (EDS) confirmed the successful encapsulation of MoS_2_ by the MWCNTs network in the composite. To measure the output force of the BMAM fabricated with different masses of MoS_2_ doping, we established a self-built experimental platform and conducted tests on the electrode membranes doped with varying quantities of MoS_2_ using an electrochemical workstation. The results revealed that the BMAM exhibited optimal mechanical performance when doped with 1.5 g of MoS_2_, with a maximum output force of 7.81 mN, an output force density of 34.36 mN/g, and a response rate of 0.09 mN/s. These performances were improved by 309%, 276%, and 175%, respectively, compared to the samples without MoS_2_ doping, with a mass-specific capacitance enhancement of 151%.

## 1. Introduction

In recent years, biomimetic artificial muscles (BMAMs) have attracted considerable attention due to their remarkable mechanical properties such as excellent flexibility, bending capability, and stretchability [1]. These characteristics enable BMAMs to simplify mechanical structures and improve energy conversion efficiency [2], providing broad prospects for their applications in flexible electronics, robotics, wearable devices, and other fields [3,4,5]. As an electroactive polymer, BMAMs are inspired by the operational principles of biological muscles. By applying an external electric field, BMAMs can mimic the deformation of natural muscles, achieve the energy conversion from electrical to mechanical energy, and generate external driving forces [6]. As an emerging technology, BMAMs not only simulate the motion characteristics of biological muscles but also possess unique performance features, including a large deformation capability, simple device structures, low cost, and environmental friendliness. Therefore, they hold significant potential for engineering applications [4,7,8]. For instance, in the field of robotics, BMAMs can be used as actuators to achieve flexible robot movements [9,10]. Additionally, they can be applied in sensitive ion sensors for detecting and monitoring changes in ion concentrations in the environment [11]. The field of smart materials can also leverage the characteristics of BMAMs to develop materials that respond to external stimuli [12]. Consequently, the study of BMAMs has become a highly regarded and popular topic in the field of materials science today.

According to previous studies, commonly used materials for electroactive polymers include synthetically derived polymers such as polyvinylidene fluoride (PVDF), polyether ether ketone (PEEK), poly(3,4-ethylenedioxythiophene) (PEDOT), and polyacrylic acid (PAA) [13,14,15,16]. However, the production and fabrication processes of these polymers result in the generation of substances harmful to human health [17]. Therefore, in this study, sodium alginate extracted from natural seaweed was chosen as the primary raw material. Sodium alginate is a natural high-molecular-weight polymer with excellent biocompatibility that can be decomposed into harmless substances within the human body [18]. Additionally, seaweed resources are abundant and renewable, making them an ideal natural, green, and sustainable material [19]. The driving performance of BMAMs is primarily influenced by the electrochemical properties of the electrode layer and the mechanical-to-electrical energy conversion process [20,21]. Hence, improving the performance of BMAMs requires exploration in two aspects: the selection of electrode membrane materials and the microstructure within the fabricated electrode membrane. In recent years, multi-walled carbon nanotubes (MWCNTs) have gained widespread attention and application in research due to their excellent conductivity, high surface area, and stable chemical characteristics [22]. The incorporation of MWCNTs significantly enhances the performance of BMAMs, including faster response speed, greater bending displacement, and higher strain rates [6]. However, MWCNTs exhibit low electric capacitance; thus, approaches involving composites with other materials can be employed to improve the electric capacitance performance of carbon nanostructures [23]. By combining the mesh-like structure formed by MWCNTs with materials possessing higher electric capacitance performance, the overall capacitance performance of BMAM can be effectively enhanced.

Molybdenum disulfide (MoS_2_) is widely utilized in the field of supercapacitors due to its high specific capacitance, fast charge–discharge rate, long cycle life, and high temperature stability [24]. In the process of preparing composite materials, MoS_2_ can effectively enhance the electrochemical performance (the ratio of material mass to its capacitance), quantifying the ability to store electrical energy of the composites. The microstructure of MoS_2_ consists of layers of sulfur atoms tightly stacked between two layers of molybdenum atoms. The presence of strong covalent bonds and weak van der Waals forces in the layered structure facilitates electron transfer between the molybdenum layers, thereby endowing it with excellent electronic conductivity [25]. Additionally, MoS_2_ is a common inorganic compound characterized by its abundant sources, low cost, and environmental friendliness [26]. Therefore, by combining the high specific capacitance, high conductivity, and high temperature stability of MoS_2_ with MWCNTs, it is possible to fabricate electrode membranes for BMAMs, aiming to enhance the electrochemical performance of the electrode membrane.

Hence, it is worthwhile to investigate the influence of different doping levels of MoS_2_ on the performance of the electrode membrane. This study aims to explore a composite electrode doped with MoS_2_/MWCNTs for the preparation of BMAMs. Through investigating the electrochemical performance of BMAMs, the correlation between their electromechanical properties and electrochemical performance will be further examined.

## 2. Materials and Methods

### 2.1. Materials

The following chemical reagents were primarily employed in the experiment: sodium alginate (AR, purity 90%, (C_6_H_7_O_6_Na)_n_, Shanghai Macklin Biochemical Co., Ltd., Shanghai, China), glycerol (CP, purity > 99%, C_3_H_8_O_3_, molar mass 92.09, China National Pharmaceutical Group Chemical Reagent Co., Ltd., Shanghai, China), MWCNTs dispersion (JCWCNDM-10, purity > 98%, CNTs diameter: 40–60 nm, CNTs content: 10%, Jiacai Technology Co., Ltd., Chengdu, China), and laboratory-prepared deionized distilled water. The main equipment employed in the preparation of BMAMs includes: a six-station magnetic stirrer (HJ-6B, manufactured by Changzhou Guowang Instrument Manufacturing Co., Ltd., Changzhou, China), a vacuum drying oven (DZF-6090BZ, Shanghai Boxun Industrial Co., Ltd., Shanghai, China), an ultrasonic cell disruptor (LC-JY92-IIN, Shanghai Wuxiang Instrumentation Co., Ltd., Shanghai, China), and an ultrasonic cleaning machine (DSA50-JY2, Fuzhou Desen Precision Machinery Co., Ltd., Fuzhou, China). The principal measuring instruments comprise an analytical balance (FA1004, Shanghai Shangping Instrument Co., Ltd., Shanghai, China), an electrochemical workstation (CHI660E, Shanghai Chenhua Instrument Co., Ltd., Shanghai, China), and a gallium ion dual-beam scanning electron microscope (FIB-SEM, TESCAN AMBER, TESCAN ORSAY HOLDING, Brno, Czech Republic).

### 2.2. Preparation of BMAMs

The preparation of BMAMs primarily involves three steps: electroactive membrane fabrication, electrode membrane formation, and manufacturing assembly. The preparation procedure is illustrated in Figure 1, and the detailed process is outlined as follows.

#### 2.2.1. Electroactive Film Fabrication

In the experiment, the heating temperature of the hexa-magnetic stirrer was set to 50 °C. A 500 mL beaker was placed on the stirrer, and 150 mL of distilled water was added to create a thermostatic water bath stirring environment. Within the large beaker, a 150 mL beaker was placed containing 100 mL of distilled water. A quantity of 1.5 g of sodium alginate was weighed on an analytical balance and gradually added to the stirrer, with a stirring speed set at 900 revolutions per minute, and thoroughly stirred for 2 h. Subsequently, 3 mL of glycerol was added and stirring continued for an additional 20 min. The resulting mixture was then subjected to defoaming treatment in an ultrasonic cleaner, which took approximately 30 min. Next, the defoamed solution was poured into a glass mold with dimensions of 100 mm × 100 mm and placed in a vacuum drying oven. The temperature of the vacuum drying oven was set to 60 °C, with a drying time of 36 h and a vacuum degree of 0. To ensure horizontal alignment, a spirit level was used to adjust the vacuum drying oven. The solution gradually dried within the vacuum drying oven, ultimately yielding an electroactive film.

#### 2.2.2. Electrode Membrane Fabrication

To achieve a homogeneous distribution of MWCNTs within a water dispersion, we employed a two-step dispersion process using an ultrasonic homogenizer. The specific procedure was as follows: the protective temperature of the ultrasonic homogenizer was set to 60 °C, with a working time of 20 min. The ultrasonic dispersion was performed for 2 s with a time interval of 5 s. Upon initiating the homogenizer, we proceeded as follows: initially, 0.5 g of sodium alginate was mixed with 80 mL of distilled water in a water bath beaker and stirred vigorously for 60 min. After complete dissolution of sodium alginate, 20 mL of the doubly dispersed MWCNTs was added. Simultaneously, 0.2 g of MoS_2_ was weighed in the beaker, which was then subjected to the same ultrasonic dispersion process as described earlier. Thus, a MoS_2_/MWCNTs hybrid solution with a doping ratio of 0.1 was obtained. Subsequently, the solution was stirred vigorously on a magnetic stirrer for 90 min, with the addition of 3 mL of glycerol, followed by an additional 20 min of stirring. The solution was then subjected to defoaming treatment in an ultrasonic cleaning machine for approximately 30 min. Finally, the defoamed solution was poured into a glass mold measuring 100 mm × 100 mm and placed inside a vacuum drying oven. The temperature of the vacuum drying oven was set to 80 °C, with a drying time of 24 h, and the vacuum level was set to 0. The solution gradually dried within the vacuum drying oven, resulting in a MoS_2_/MWCNTs electrode film with a doping mass of 0.2 g. Under the same process conditions, we prepared MoS_2_/MWCNTs electrode films with doping masses of 0.5 g, 1 g, 1.5 g, and 2 g using the same ultrasonic oscillation adsorption method.

#### 2.2.3. Assembly of the BMAM

The prepared electrode film was placed on a clean filter paper and then subjected to thermal pressing in a pneumatic hot-press machine. The preheating temperature of the hot press machine was set to 40 °C, and the hot pressing pressure was set to 0 MPa for a duration of 15 min. Subsequently, a piezoelectric film was aligned and placed above the electrode film, and another clean filter paper was used for additional thermal pressing. The temperature for this second pressing step was also set to 40 °C, with a hot-pressing pressure of 0.1 MPa and a duration of 15 min. Next, the same procedure was followed to place a second electrode film above the piezoelectric film and perform hot pressing using the same parameters, with a duration of 15 min. In this way, we successfully fabricated a BMAM. The fabricated BMAM was then cut into specimens with dimensions of 35 mm × 8 mm for experimental purposes.

### 2.3. Testing Methods

This study employed various experimental instruments to systematically test and measure the specimens. Firstly, a field emission scanning electron microscope (FIB-SEM) was utilized to conduct a detailed observation of the surface morphology of the samples, accompanied by energy-dispersive spectroscopy (EDS) analysis to acquire information on the chemical composition. Secondly, in order to investigate the electrochemical performance of the samples, an electrochemical workstation (CHI660D) was selected for further experiments. By employing cyclic voltammetry, the specific capacitance of the samples was obtained. Figure 2c illustrates the measurement method for the bending force. According to the experimental results, it can be observed that the BMAM is in a horizontal state when no voltage is applied (as shown in Figure 2a). Subsequently, we applied a 4V DC voltage to the biomimetic artificial muscle, and as a result of this voltage being applied between the clamps, the muscle underwent a relative bending deformation with respect to its own structure. Furthermore, this bending occurred in the direction towards the positive pole (as depicted in Figure 2b). Remarkably, the BMAM exhibited deviation towards their anode when electrically stimulated through electrode membranes. Simultaneously, the generated bending force was monitored in real time through high-precision analysis using a weighing scale for sampling purposes. To ensure data reliability, all experimental data provided in this study are based on the average values obtained from three samples, mitigating the influence of experimental errors. Through the application of these comprehensive experimental techniques, we were able to gain a comprehensive understanding of the morphology, mechanical properties, and electrochemical performance of the samples, thereby providing reliable data support for subsequent research endeavors.

## 3. Results and Discussion

### 3.1. Analysis of Scanning Electron Microscope Results

To validate the adsorption effect of ultrasonic oscillation on MoS_2_ particles on MWCNTs, we employed transmission electron microscopy to scan two groups of electrode films with MoS_2_ doping masses of 0 g and 1 g, respectively, and detected the content of Mo, O, and C elements. Quantitative information regarding the elemental composition on the sample surface was obtained through the scanning tests. Table 1 presents the results of the scanning tests and energy-dispersive X-ray spectroscopy (EDS) analysis. By analyzing the spectra and peak intensities, we were able to determine the relative content of Mo, O, and C elements and observe their distribution on the sample surface (as shown in Figure 3). These data will contribute to the validation of the adsorption effect of ultrasonic oscillation on MoS_2_ particles on MWCNTs and provide crucial experimental evidence for further investigations into their applications.

Table 1 presents the mass percentage and atomic percentage of Mo in the two electrode membranes. Figure 3a indicates that Mo was not detected, whereas Figure 3b shows that it was detected in this experiment, with a mass percentage of 26.46% and an atomic percentage of 4.35%. Based on these detection results, we can conclude that MoS_2_ was successfully incorporated into the interior of the BMAM electrode membrane, and the ultrasonic oscillation method was demonstrated to embed MoS_2_ onto the surface of MWCNTs. To obtain a clearer observation of the surface morphology of the BMAM electrode membrane and the distribution of MoS_2_ on the membrane surface, high-magnification field emission scanning electron microscopy (FIB-SEM) was employed for examination.

The surface morphology of electrode films with different doping masses of MoS_2_ was observed using field emission scanning electron microscopy (FIB-SEM). Figure 4 presents the microstructure of electrode films with varying MoS_2_ doping masses, with the left image magnified 20,000 times and the right image magnified 50,000 times. The observation results reveal significant differences in the microstructure within electrode films of different MoS_2_ doping masses. Combining the results of the energy spectrum analysis, the following conclusions can be drawn: when the doping mass of MoS_2_ is 0 g, there are no MoS_2_ particles present on the surface of the electrode film; when the doping mass of MoS_2_ ranges from 0.2 g to 0.5 g, a small amount of MoS_2_ is adsorbed on the surface of the electrode film; when the doping mass of MoS_2_ ranges from 0.5 g to 1.5 g, a greater amount of MoS_2_ is adsorbed on the surface of the electrode film; when the doping mass of MoS_2_ ranges from 1.5 g to 2 g, there is no significant increase in the amount of adsorbed MoS_2_ on the surface of the electrode film. This indicates that the network structure formed by MWCNTs has a certain limit to the adsorption of MoS_2_. When the doping mass of MoS_2_ exceeds 2 g, the adsorption capacity of this network structure tends to saturate, and it cannot adsorb more MoS_2_ particles.

### 3.2. Analysis of Output Force Testing Results

In the experimental process, we measured the output force of BMAM doped with different quantities of MoS_2_ using a self-built experimental platform. The duration of the experiment was 300 s, during which we collected 600 data samples. These data were subjected to polynomial fitting using Origin 2021 9.8 software. Through the analysis of the fitted data, we obtained the line graph depicting the output force of BMAM doped with different quantities of MoS_2_ over the specified time period (as shown in Figure 5a). From the graph, it can be observed that the BMAM reached their maximum output force in approximately 60 s. Figure 5a illustrates the relationship between output force magnitude and time. The slope at each point on the curve represents the rate of change in the output force over time, which we refer to as the output force response rate. Using the Origin2021 software, the maximum average slope of the curve was determined, representing the response rate for the given doped mass. Furthermore, we calculated the ratio of the maximum output force to the mass of the BMAM, resulting in the output force density, which represents the maximum output force provided by the BMAM per unit mass. By plotting the curves of the output force density and the response rate (as shown in Figure 5b), we were able to observe the variations under different doping ratios.

Figure 5a shows the influence of MoS_2_ doping with different mass fractions on the output force of the BMAM. The study revealed that the doping of MoS_2_ can enhance the maximum output force of the BMAM within the electrode membrane. As the mass fraction of MoS_2_ doping varies, the maximum output force of the BMAM exhibits an initially increasing and then decreasing trend. When the mass fraction of MoS_2_ doping reaches 1.5 g, the maximum output force reaches 7.81 mN, representing a 309% improvement compared to undoped MoS_2_. However, when the mass fraction of MoS_2_ doping increases to 2 g, the output force decreases to 6.13 mN. To provide a more intuitive representation of the performance trend in the BMAM, we plotted the variations in output force density and response rate of the MoS_2_-doped artificial muscle with different mass fractions, as shown in Figure 5b. The trends observed in the output force density and response rate changes depicted in Figure 5b are generally consistent with the trend in maximum output force variation shown in Figure 5a, exhibiting an initially increasing and then decreasing trend. When the mass fraction of MoS_2_ doping is 1.5 g, the output force density reaches 34.36 mN/g, and the response rate reaches 0.09 mN/s, representing a 276% and 175% improvement in performance, respectively, compared to undoped MoS_2_. However, when the mass fraction of MoS_2_ doping increases to 2 g, the output force density decreases to 26.81 mN/g, and the response rate decreases to 0.083 mN/s. This phenomenon is attributed to the adsorption capacity of MWCNTs within the electrode membrane having reached its carrying limit. The research results indicate that, with the increase in the MoS_2_ doping mass fraction, further increasing MoS_2_ content cannot enhance the mechanical performance of the BMAM.

### 3.3. Analysis of Electrochemical Testing Results

Based on previous research findings [27,28], doping electrode membranes with high-electrochemical-performance materials can significantly enhance the mechanical properties of BMAMs. Molybdenum disulfide (MoS_2_), a transition metal sulfide, possesses a high specific surface area and conductivity, making it widely employed in supercapacitors and recognized as an excellent energy storage material. To thoroughly investigate the influence of different mass loadings of MoS_2_ doping on the electromechanical performance of the electrode membrane, we employed an electrochemical workstation to measure the current–voltage (CV) characteristic curves of MoS_2_ electrode membranes with various mass loadings (0 g, 0.2 g, 0.5 g, 1 g, 1.5 g, and 2 g). During the experiment, we set the scan rate to 50 mV/s to examine the impact of different mass loadings of MoS_2_ doping on the specific capacitance of the electrode membrane.

Figure 6a shows the CV curves of MoS_2_ electrode films with different doping masses. The specific capacitance values of the electrode films were calculated by measuring the area enclosed by the curves, considering the sample mass, scan rate, and potential window. The specific capacitance values obtained were 31.69 mF/g, 36.79 mF/g, 40.43 mF/g, 43.42 mF/g, 47.9 mF/g, and 48.8 mF/g. It can be clearly observed from the graph that the addition of MoS_2_ to the electrode film effectively enhances its specific capacitance. To visually demonstrate the influence of MoS_2_ doping with different masses on the mass specific capacitance, we plotted Figure 6b, which clearly illustrates the rate of change in the specific capacitance. The experimental results indicate that the specific capacitance of the electrode film gradually increases with increasing MoS_2_ doping. When the doping mass is in the range of 0~0.5 g, the specific capacitance of the electrode film increases rapidly. However, in the range of 0.5~1.5 g doping mass, although the specific capacitance continues to increase, the growth rate decreases. When the doping mass reaches 1.5~2 g, the increase in specific capacitance becomes small and tends to stabilize. At a doping mass of 2 g, the electrode film reaches its maximum specific capacitance of 48.8 mF/g, which represents a 154% improvement compared to the undoped MoS_2_ electrode film.

To investigate the influence of the electrochemical properties of electrode films on the performance of output force, we analyzed the variation patterns of specific capacitance, output power density, and output power response rate by doping MoS_2_ electrode films with different masses. Figure 7a reveals that the specific capacitance and output power density of the electrode films exhibit a positive correlation with the doping mass of MoS_2_, albeit with different growth rates. Within the doping mass range of 0 g to 1 g, the growth rates of both parameters are relatively similar. However, when the doping mass reaches 1 g to 2 g, the growth rate of specific capacitance becomes slower, while the output power density shows an increasing–decreasing trend. Figure 7b demonstrates that the specific capacitance and output power response rate of the electrode films also exhibit an overall positive correlation with the doping mass of MoS_2_. Within the doping mass range of 0 g to 0.2 g, both parameters increase as the doping mass increases, with similar growth rates. As the doping mass ranges from 0.2 g to 0.5 g, the growth rate of the output power response rate significantly increases, surpassing that of the specific capacitance. Meanwhile, within the doping mass range of 0.5 g to 2 g, the growth rate of specific capacitance steadily rises, while the output power response rate remains relatively unchanged, especially with a declining trend between 1.5 g and 2 g of doping mass.

In summary, by doping MoS_2_ in the electrode membrane, its specific capacitance can be enhanced. This phenomenon is attributed to the enormous specific surface area of MoS_2_, which enables it to accommodate a large number of free electrons, thereby significantly improving the electron storage capacity of the electrode membrane. Under the influence of a direct current stabilizing power supply, the internal layer of the biomimetic artificial muscle electrode can accommodate a greater number of free electrons, resulting in a higher electric field generated across the electroactive film. This, in turn, enhances the mechanical performance of the biomimetic artificial muscle, including higher output force density and faster response rate. However, when the doping mass of MoS_2_ reaches 1.5 g, it reaches the adsorption limit of MWCNTs and cannot adsorb more MoS_2_. Therefore, no significant improvement in the mechanical performance of the biomimetic artificial muscle was observed when the doping mass of MoS_2_ exceeded 1.5 g.

## 4. Conclusions

In conclusion, this study utilized the ultrasonic oscillation method to prepare a BMAM electrode membrane by incorporating varying masses of MoS_2_, resulting in the formation of MoS_2_/MWCNTs composite electrode membranes. Energy-dispersive spectroscopy analysis revealed that when the mass of MoS_2_ dopant was 1 g, the mass percentage of Mo element was 26.46% with an atomic percentage of 4.35%. The test results indicated that the ultrasonic oscillation method effectively adsorbed MoS_2_ particles onto MWCNTs. The surface morphology of the composite electrode membrane was observed through scanning electron microscopy. Due to the inherent capacity limitation of the network structure formed by MWCNTs, the BMAM exhibited optimal output force performance when the mass of MoS_2_ dopant was 1.5 g. The maximum output force was 7.81 mN, the output force density was 34.36 mN/g, and the response rate was 0.09 mN/s, representing improvements of 309%, 276%, and 175%, respectively, compared to the sample without MoS_2_ doping. The incorporation of MoS_2_ significantly increased the specific surface area of the composite electrode membrane, thereby enhancing its electrochemical performance and specific capacitance. The specific capacitance of the electrode membrane exhibited a positive correlation with the mechanical performance of the BMAM. Although the electrode membrane achieved its maximum specific capacitance of 48.8 mF/g at a doping mass of 2 g, a 154% increase compared to the undoped MoS_2_ electrode membrane, and surpassing the value of 47.9 mF/g obtained at a doping mass of 1.5 g, no significant improvement in the mechanical performance of the biomimetic artificial muscle was observed when the doping mass of MoS_2_ exceeded 1.5 g.

## Figures and Tables

**Figure 1 polymers-15-03535-f001:**
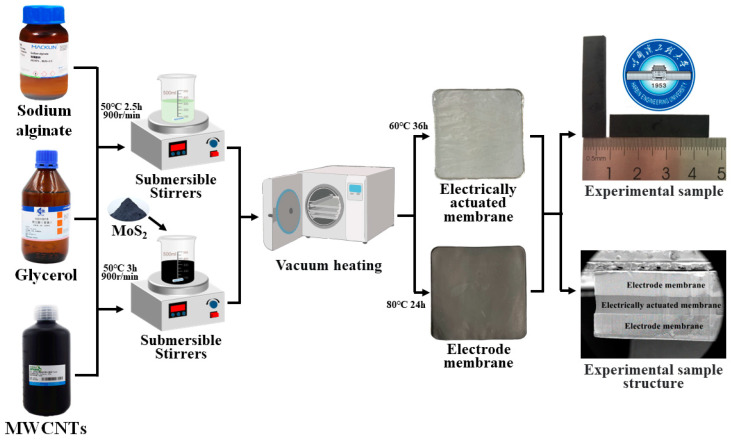
Process flowchart for the fabrication of BMAMs.

**Figure 2 polymers-15-03535-f002:**
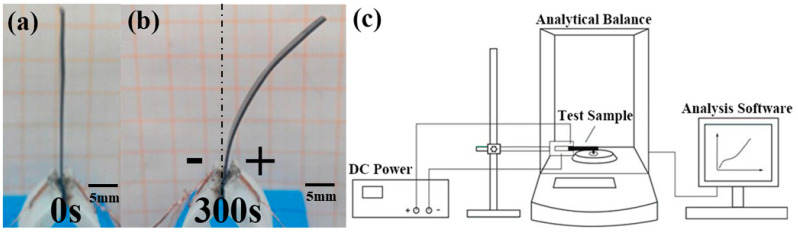
The measurement method for the bending force. (**a**) Initial state of BMAM; (**b**) BMAM exhibits a bending phenomenon under the influence of DC voltage; (**c**) schematic diagram of the experimental platform.

**Figure 3 polymers-15-03535-f003:**
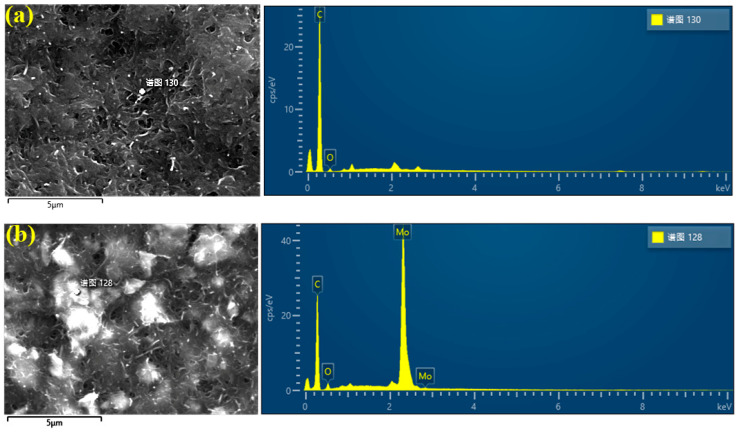
EDS results of electrode films with MoS_2_. (**a**) EDS testing results of MoS_2_ with 0 g doping mass; (**b**) EDS testing results of MoS_2_ with 1 g doping mass. (The “谱图” in the figure means “Spectrum”.)

**Figure 4 polymers-15-03535-f004:**
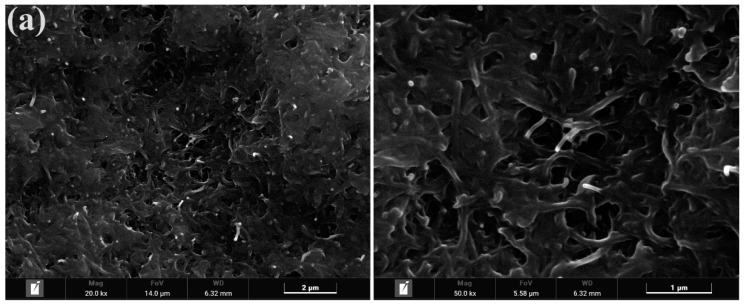
FIB-SEM images of electrode membranes doped with different mass fractions of MoS_2_, with a magnification of 20,000 times on the left side and 50,000 times on the right side; (**a**) 0 g; (**b**) 0.2 g; (**c**) 0.5 g; (**d**) 1 g; (**e**) 1.5 g; (**f**) 2 g.

**Figure 5 polymers-15-03535-f005:**
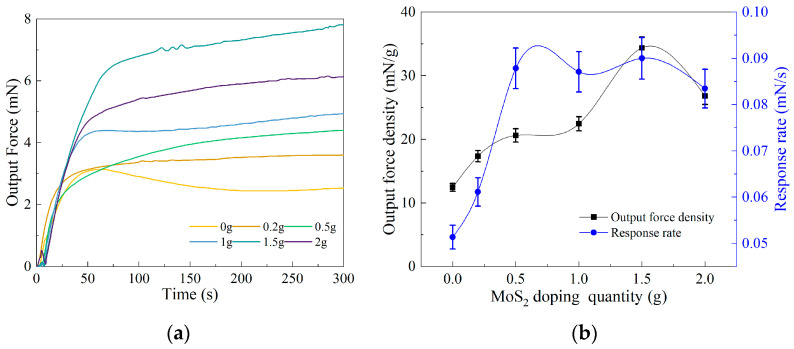
Output force testing results of BMAM: (**a**) The output force of BMAM doped with different quantities of MoS_2_; (**b**) the output force density and the response rate of BMAM.

**Figure 6 polymers-15-03535-f006:**
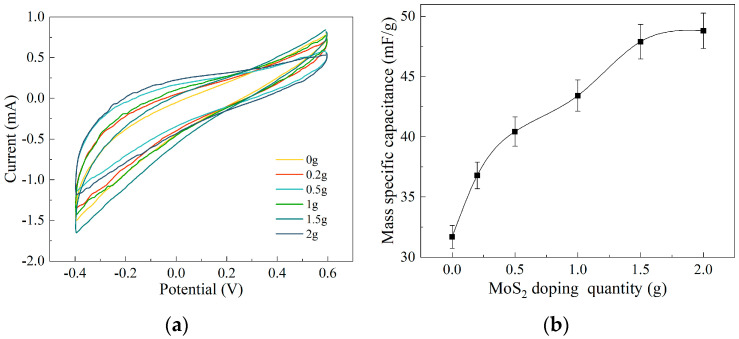
Electrochemical testing results of electrode membranes: (**a**) CV curve of electrode film doped with different masses of MoS_2_; (**b**) MoS_2_ doping with different masses on the mass specific capacitance.

**Figure 7 polymers-15-03535-f007:**
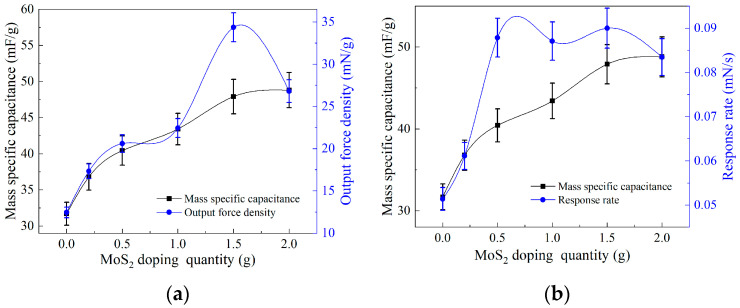
Relationship between specific capacitance and mechanical properties of electrode films doped with different masses of MoS_2_: (**a**) Relationship between specific capacitance and output force density; (**b**) relationship between specific capacitance and output force response rate.

**Table 1 polymers-15-03535-t001:** Mass percentage and atomic percentage of Mo in two electrode membranes.

Element	Position 1		Position 2	
	Mass (%)	Atomicity (%)	Mass (%)	Atomicity (%)
C	96.7	97.02	70.58	92.73
O	3.93	2.98	2.96	2.92
Mo	0	0	26.46	4.35

## Data Availability

The data used for the research described in this manuscript are available upon request.

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
