# Peer review of "Bioinspired Artificial Muscles Based on Sodium Alginate-Wrapped Multi-Walled Carbon Nanotubes and Molybdenum Disulfide Composite Electrode Membrane"

_polymers, 2023, doi:10.3390/polym15173535_

Round 1

Reviewer 1 Report

MS „Y. Ji, K. Wang and G. Zhao: Bioinspired Artificial Muscles based on... (polymers-2542752)” reports results on characterization of biomimetic artificial muscle (BMAM) preparation based on sodium alginate, MWCNT and MoS2.

The topic is very important in academic and industrial laboratories, and, consequently, can be interesting for potential readers. The experimental design is correct, and the authors used sophisticated technologies in addition to routine laboratory procedures. The structure of the MS itself is not bad, however, the data were not presented in a convincing way. Based on the importance of the topic I suggest publication after major revision addressing the issues listed below.

 Major concerns:

1.       What kind of deformation is presented by the sample, how is the directionality of the deformation assured in the sample? Is there any directionality? What kind of stress is measured? Shear stress or tensile stress?

2.       The demonstration of the reproducibility of the sample response is completely missing, however, it is important for any practical application. Also, statistical analysis is missing completely. Is there a hysteresis in the deformation? I would be surprised not to have it. How reproducible is the single sample measurement, and what is the variability of responses of different preparations?

3.       Without basic statistical evaluation I doubt any serious conclusion of figures like Figs. 5 and 7.

4.       It is not clear how the response rate was determined? It is stated in L.236: “Since we employed linear fitting to represent the output force, the slope of the curve can be approximated as the response rate of the BMAM, which is obtained by dividing the maximum output force by the time taken to reach it. 

-       - How the time for reaching the maximum output was determined?

-       - Has the slope anything to do with the magnitude of the maximum output force?

-       - Actually, the slope is changing with time. Would not be the initial slope an important (more exact) parameter?

   5.           I disagree with this kind of data presentation as the wavy point-by-point connection lines shown in Figs. 5b, 6b, 7a,b. My general view is that points come from the measurements, lines come from the model. However, no model is presented here for drawing the curves. It is specially confusing here because no error bars are indicated.

Just an example: if the measurement with 1.5g MoS2 for output force density (Fig. 7a) is a mistake (note, no error calculation is indicated), no difference in tendency of the two parameters can be concluded. Even more, the conclusion presented in the MS does not hold.

Other concerns:

-         L.9.: “Using the high molecular polymer sodium alginate...” – „high molecular polymer”?

-         L.16.: “… energy-dispersive spectroscopy …” – of course this is X-ray spectroscopy, as defined correctly in L.182. However, this appeared in Ls.16, 157. – Please, specify it at its first appearance.

-         L.17.: “… MoS2 within the MWCNTs...” – “within?”

-         L.65, 66, 68.: “… capacitance…” I would specify it is “electric capacitance” at first appearance, because one could realize it later only) L.71 (may be).

-         L.72.:  What is the meaning of “… ideal electrochemically-enhanced material…”?

-         L.91.:  “molecular weight” does not exist. “molar mass” is correct.

-         L.116.:  “… a vacuum degree of 0…” – is this value the instrumental setting?

-         L.144.:  “… the hot pressing pressure was set to 0 MPa…” – see comment to L.116.

-         L.155.: FE-SEM, FESEM or just SEM?

-         L.184.: “… their distribution on the sample surface (as shown in Figure 3)...” – in Fig. 3 I don’t see the surface distriution of these elements.

-         On Figs. 5,6,7 is it not better “doping quantity” instead of “doping quality”?

I am not native English speaker, I am not qualified to evaluate the English quality of the MS. However, I found the readability of the MS good, no issue is detected.

Suggestion: major revision.

Author Response

Dear respected reviewer, it is an utmost honor to have the opportunity to engage in a scholarly dialogue with an esteemed expert and scholar like yourself. I sincerely appreciate the professional and academic insights you have provided on the manuscript. Your diligent efforts are genuinely acknowledged and deeply valued.As a first-time contributor to the academic discourse, I am well aware of the arduous and enduring journey of academia. Each of your comments serves as a guiding beacon, directing me towards the path of progress.

Upon careful reflection on your recommendations, I have made several revisions to the paper. Once again, I extend my heartfelt gratitude for your invaluable feedback and your patient consideration. Thank you sincerely for your time and expertise.

Reviewer 2 Report

1. Please provide Figure 1 in more detail. For example, the duration of using the stirrer or the amount and time of heat, etc

2. Mention the specifications of the devices used.

3. The results have been well presented, however, there has not been a proper discussion with the latest published scientific articles. Please do more comparison and discussion between the results of the design with the last similar cases presented.

Author Response

(The authors gave the same response as above.)

Round 2

Reviewer 1 Report

MS „Y. Ji, K. Wang and G. Zhao: Bioinspired Artificial Muscles based on... (polymers-2542752-peer-review-v2)” is a revised version of MS submitted to the journal by the same authors with the same title.   

Most of the reviewer’s comments are addressed and, in my opinion, the quality of the MS improved significantly. Nevertheless, few, but important questions remained. I still would like to see the MS before publishing it after the second revision.

1)      Regarding my first question in first review (“What kind of deformation is presented by the sample, how is the directionality of the deformation assured in the sample? Is there any directionality? What kind of stress is measured? Shear stress or tensile stress?”)

Please, comment in the MS shortly what is the physical phenomenon and structural requirements behind the effect of DC voltage on the bending? Also, what structural arrangement determines the directionality of the deformation. If these are known.

2)      Regarding question 2 („It is not clear how the response rate was determined?): If it is the “the maximum average slope” as stated, how the authors determined the time intervall that should be averaged? Because it should be dependent on the time interval.

3)      Regarding question 5: My concern is not with „solid thick and thin lines”. I disagree with piont-by-boint connection of measured data without model calculations. Specially with curved lines. Here, after presenting the error bars in the revised version data became more convincing to me. Anyway, if the journal accepts this kind of presentation I am not aganst it.

Other comments:

-         L.17: “encapsulation of MoS2 by MWCNTs” (same in the first version of the MS) – Does „encapsulation” mean that MoS2 is inside MWCNT? Probably not.

-         L.73: I still don’t understand what does “electrochemical performance” mean. What parameter do the authors mean on „performance”?

-         L.171: „... 4V direct current to the biomimetic ...” – Sorry, „4V” is voltage and not current. By the way, what is the cause of the deformation? Voltage or current?

-         L.195: “… observe their distribution on the sample...” – Sorry, I still don’t see the „distribution”. I see the sample content, but I don’t see distribution – I mean element „mapping”.

-         Please, put reference lines indicating distances in Figs. 2a and b.

-         Finally I want to pay attention to correct writing of chemical structures (using subscripts).

Finally, because of the importance of the topic and relevance in the field, I suggest publication of the MS.

However, because I still have major concerns my suggestion is “accept with major revision”.

Author Response

Dear Reviewers, It is a great honor to have the opportunity to engage in a scholarly dialogue with respected experts and scholars such as yourself. I sincerely appreciate the professional and scholarly insights you have provided on the manuscript. Your diligence is truly recognized and highly valued. As a first-time contributor to the scholarly discourse, I am well aware of the difficult and enduring journey of the academy. Each of your comments has been a guiding light that has guided me along the path of progress.

After careful consideration of your suggestions, I have made several changes to the document. Once again, I would like to thank you for your valuable feedback and patience. I sincerely appreciate your time and expertise.

Round 3

Reviewer 1 Report

MS „Y. Ji, K. Wang and G. Zhao: Bioinspired Artificial Muscles based on... (polymers-2542752-peer-review-v3) is a second revised version of MS submitted to the journal by the same authors with the same title.   

Most of the reviewer’s comments are addressed and, in my opinion, the quality of the MS improved significantly again, it is ripened enough for publication.

I still suggest a few changes, perhaps due to a little misunderstanding in our discussions.

I would like to ask the authors to reconsider the following suggestions for finalizing the MS as follows.

Minor comments:

1)  L.17:  „... encapsulation of MoS2 by MWCNTs.

Perhaps more correct: „... encapsulation of MoS2 by MWCNTs network in the composite”.

2) Please difine electrochemical performance ” in the text.

E.g. in L.73: „... enhance the electrochemical performance of the composites.

May be: „... enhance the electrochemical performance (the ratio of material mass to its capacitance), quantifying the ability to store electrical energy of the composites.

3) L.171: „... a 4V DC voltage to the biomimetic artificial muscle, and as a result of this current being...”

“…this current...”? may be “…this voltage…”

Final comment: accept the MS after minor modifications.

Footnote for the authors:

 1)      You don’t have to explain your work so deeply to the reviewers, because the reviewers probably understand the material, they are experts in the field. You should make it clear to possible readers.

2)      The readers (and reviewers) do not care about your PhD and experience. The reader is interested only in the message and the quality of the paper.

3)      I completely disagree with the method of calculating the average slope. It appears that the curves reflect heterogeneous kinetics, which can’t be averaged out. I don’t understand why Max(dOutput Force/dTime) would not be a good parameter.

4)      I know that even respected journals usually accept figures in which points are connected point-by-point. However, I am convinced that this way of presentation is completely wrong (and not nice). Especially the wavy lines in Figs. 5b, 6b, 7a,b. Points come from measurement, lines come from the model.

5)      I feel controversy between the statement in Ls.158-159 vs. Fig. 2a,b. How can we compare 3.5mm x 8mmm sample size with 5mm scaling?

Author Response

(The authors gave the same response as above.)
